# Fire Behavior of Wood-Based Composite Materials

**DOI:** 10.3390/polym13244352

**Published:** 2021-12-13

**Authors:** Juliana Sally Renner, Rhoda Afriyie Mensah, Lin Jiang, Qiang Xu, Oisik Das, Filippo Berto

**Affiliations:** 1School of Mechanical Engineering, Nanjing University of Science and Technology, Nanjing 210094, China; rennerjs@yahoo.com; 2The Division of Material Science, Department of Engineering Sciences and Mathematics, Luleå University of Technology, 97187 Luleå, Sweden; rhoda.afriyie.mensah@ltu.se (R.A.M.); oisik.das@ltu.se (O.D.); 3Department of Mechanical and Industrial Engineering, Norwegian University of Science and Technology (NTNU), 7491 Trondheim, Norway

**Keywords:** wood-based composites, wood plastic composites, flammability properties, mechanical properties, flame retardants

## Abstract

Wood-based composites such as wood plastic composites (WPC) are emerging as a sustainable and excellent performance materials consisting of wood reinforced with polymer matrix with a variety of applications in construction industries. In this context, wood-based composite materials used in construction industries have witnessed a vigorous growth, leading to a great production activity. However, the main setbacks are their high flammability during fires. To address this issue, flame retardants are utilized to improve the performance of fire properties as well as the flame retardancy of WPC material. In this review, flame retardants employed during manufacturing process with their mechanical properties designed to achieve an enhanced flame retardancy were examined. The addition of flame retardants and manufacturing techniques applied were found to be an optimum condition to improve fire resistance and mechanical properties. The review focuses on the manufacturing techniques, applications, mechanical properties and flammability studies of wood fiber/flour polymer/plastics composites materials. Various flame retardant of WPCs and summary of future prospects were also highlighted.

## 1. Introduction

In recent times, key industries such as construction and building mainly rely on innovative materials which are innocuous toward the environment. Constant efforts have been made to develop new products that impart sustainability. One of the most critical inventions growing from wood products are wood-based composite materials. The distinctive properties of wood-based composites allow for the creation of products that are sustainable and environmentally friendly [1]. It is worth noting that wood is commonly used in the construction sector and other fields. However, more advancements are required to expand its usefulness.

Generally, wood-based composite (WBC) is manufactured from varieties of derivative materials made from wood products such as timber or lumber processed into boards, wood waste or wood chips [2] which can be engineered as composites products. In actual fact, one-half volume of the logs converted as timber or lumber are released [3], the remainder is considered to be waste. The amount of wood chips, bark, slabs, edging, sawdust, planer shavings, plywood trim etc. during processing can now be converted into useful application. Using these wood products as composites can be beneficial as it will reduce the quantity of wood waste that are incinerated or buried during disposal. Consequently, wood composite products can also be manufactured with fibers as well as using materials containing lignin with the application of adhesive or resins to enable the combined materials integrate well and therefore enhancing their performance [4].

The unique innate properties of wood-based composites, such as thermal insulation properties, toughness, aesthetic appeal, light weight, resistance to corrosion, etc., has deemed these materials very useful in various fields [5,6]. As a result, use of WBC materials is rapidly increasing in industries such as automobile, construction, and electronics; however, it is largely used in construction industries [7]. In addition, the usage of WBC materials presents a cost-effective alternative compared to the use of conventional materials. Presently, construction establishments are determined to indestructibly motivate an extension of sustainability, weight reduction and a wider advancement in thermal insulation properties. Thus, wood-based composites (WBC) facilitate improvements in the environmental performance of wood materials [8,9].

The mainstream applications for WBC in industrial sectors gaining a widespread growth are packaging materials, furniture, pallets, panels, structural framework for architectural designs, bridges, etc. [10]. However, the largest market for wood-based composites (WBC) is both residential and commercial building applications. About 95% of residential housing in the United States are built with wood-based composite materials and this industry also uses them for non-structural applications [11]. Due to the depletion of forest resources, the development of new advancements and technologies for wood-based composites recently, traditional materials for housing construction have been substituted progressively by engineered wood products (EWPs). Solid wood floors, wall diaphragms, paneling and ceiling lining have instead been sheathed with structural composite panels such as plywood and OSB.

Wood composites are generally elucidated as an array of products with a combination of wood elements held together by a binder [12]. A major advantage of wood composites is that they can be designed to performance a specific requirement at different varieties, geometries and measurements: thicknesses, grades and sizes. Wood composites are manufactured to take advantage of the natural strength characteristics of wood (and sometimes this results in a greater structural strength and stability than regular wood) [13,14]. On the other hand, wood composites also have limitations; they require more primary energy for processing and manufacturing when compared to solid lumber. Long usage for outdoor decking is not suitable especially in hot weather as it may warp under high temperatures. However, nanotechnology can be used to improve the quality of wood-based composites to fulfill the increasing demand for existing products and for new products to be used in new applications [15,16].

The basic element for wood-based composites is fiber, wood particles, flakes, veneers, laminated or lumber, although they come in varieties of forms in sizes and shapes [17]. Figure 1 displays the basic wood elements for wood composites.

Again, varieties of wood products are appropriate and can effectively be used for wood-based composites including wood defects such as knots. These wood elements use adhesives to bind them in a processing technique to form panels such as fiberboards, plywood, strand boards, particleboard, etc. [19]. Additionally, wood-based composites include a wide range of different derivative of wood products all of which they are created by binding the strands, fibers or wood boards together with plastics and some additives to improve its flammability and mechanical properties [20].

These materials are created to form the union of two or more components to obtain a composite material suitable to improve the material’s property. The material which contains the majority phase is termed as the matrix, which can be produced involving metallic, ceramic or mineral and organic or polymeric [21]. Metallic composites such as aluminum/boron fibers and aluminum/carbon fibers are used to form a fiber reinforced metallic composite [22]. Among the mineral composites are concretes such as sand, cement, and additives, carbon-carbon fibers and ceramic fibers [23]. Lastly, the organic composites which includes resins and cellulose fibers, glass fibers including others are used with various techniques to form WBC products. Table 1 shows typical composites generally used in combination with wood to form wood-based composites.

The concept of wood-based composites is the formulation and combination of raw materials, usually natural or synthetic to develop a finished product to satisfy a specific function in the concern industry, especially in the construction sector [24,25,26]. Composite materials are sometimes called reinforcement because of its arrangements and fillers implanted in a matrix [27]. The matrix enables cohesion and orientation of the load and ensures the transmission of load stresses to which the composite is subjected. The matrix, which is usually thermoplastic, thermosetting and elastomer, is linked to the reinforcing fibers distributing restraints, enabling the chemical resistance of the structure to give a desired geometry to the final product [28].

To take advantage of waste generated by wood industries and preserve the landfill which adversely affects the surroundings/environment, a suitable solution is to combine or blend the polymeric materials with the wood waste to form a composite [29]. This in turn enables the manufacturing of wood plastic composites (WPC) for a wide range of structural and nonstructural applications in various engineering field [30,31]. The mixing or blending of the composite materials may be achieved by using conventional polymer processing techniques which includes compression, injection and extrusion processes. However, the manufacturing of WPC generally involves two steps: compounding of the wood fiber/flour with melted plastics to produce wood plastic composite and lastly heating and compressing the composite materials into a desirable shape to attain the finish product.

Mechanical properties of WPC are of importance for their intended applications because product strength serves as a primary criterion for product specification and long-term service. Like other composite materials, mechanical properties of WPC frequently depend on several factors; weight/load fraction or ratio of the components, type of additive used and manufacturing techniques employed. Generally, WPCs often contain a certain amount of additives for enhancement and enable ease of manufacturing processes, thus achieving higher performance of the resulting products for its intended purpose. Additives such as coupling agents, stabilizers, lubricants, pigments, etc. greatly influence the overall performance of WPC. Flame retardants are important class of additives added into wood plastic composite (WPC) to enhance the fire performance of the WPC, especially in structural applications [32,33,34].

The reinforcement provides and contributes to the mechanical strength; thus, the tensile stress and the rigidity. Whiles the filler is said to be designated by any substance when added to the base polymer, making it possible to influence the mechanical and thermal properties to improve the surface of the composite materials [35].

Researchers have taken a keen interest lately in the production of wood composites especially with plastic due to their diverse applications and physical properties [36,37]. Wood plastic composite is very sturdy and exhibits duplicate advanced structural insulation properties as solid wood with excellent moisture resistance and decay. Plastics such as polypropylene, polyvinyl chloride, polystyrene, polylactic acid, and polyethylene are the most common materials among others suitable for wood composite production which will be discussed in this review.

Over the decades, many review articles on WPC have been published [38,39]. They focused on modern production techniques, chemical modification, fire retardant, and types of WPC materials used for a variety of applications. This review is intended to review wood-based composite materials (WPC) its production, properties, applications, fire behavior of the composite materials, wood as reinforcement agent, fire retardant and its mechanism, new prospects toward construction, and lastly the developing trend of wood-based composites in the future.

## 2. Wood Polymer Composites (WPC) and Manufacturing

Wood polymer composites, otherwise called wood plastic composites (WPC), are composite materials made from a mixture of wood flour/wood fibers and with some additives or thermoplastic resins, such as polypropylene (PP), polyethylene (PE) and polyvinyl chloride (PVC) or polylactic acid (PLA), and to a minor degree biopolymer including others that serves as a matrix [39]. They are also materials in which wood is filled with monomers that are polymerized in the wood to modify the material which facilitate optimal processing condition for a specific application. In addition, WPC can also contain other ligno-cellulosic and some inorganic filler materials. WPC can be manufactured from environmentally friendly materials such as wood chips, wood waste, wood fibers, unused natural resources, and recycled thermoplastic resins [40,41,42].

Due to many excellent properties of WPC materials that result from high durability, specific strength, resistance to wear, lightness, enhanced mechanical properties to a wider and greater sustainability for a growing number of applications in various engineering field. Manufacturing of WPC materials have highly dominated in building industry due to their excellent molding performance and good texture similar to that of solid wood. The manufacturing and major application of WPC materials are for exterior deck flooring, railing, fencing, landscaping, cladding, window and door frames, furniture, etc. [43].

WPC has advantages over other wood composites due to its high resistance to decay, though it does absorb some water into the wood fibers embedded within the material [44]; however, it can be enhanced by acetylation treatment to have a greater mechanical performance [45]. It has good workability and ability to shape and framed to meet any desired standards using conventional woodworking tools. WPC materials can be oriented to form strong arching curvatures and fastens well with good grip when screwed with nails than in solid wood [46]. Another advantage is its aesthetic, with excellent finish which does not need paint as it comes in a variety of colors but are widely available in gray and earthen brown or blends of coffee color.

However, many investigations and technologies continue to mature in manufacturing processes to improve the performance and for a wider use of applications of WPC. The production of WPC is by thoroughly mixing wood flour or wood fiber and heated thermoplastic resin. The mixing or blending of the polymer and wood flour involves the ratio of wood to plastic in the composite which determine the melt flow index of the WPC. The most common technique used for the production is extrusion or compression molding, although injection molding can also be used. Heat and shear forces are applied to a polymer within the barrel of an extruder to blend polymer with the wood fiber, additives to make pellets of compounded material and extrude solid profiles, flat sheets or hollow sectioned profiles to the required dimension according to a specific application. A wide variety of injection molded parts are produced from automotive door panels and extend to other engineering fields.

WPC may be manufactured from either natural or recycled thermoplastic including polypropylene (PP), polystyrene (PS), polylactic acid (PLA), high- and low-density polyethylene (HDPE and LDPE), and acrylonitrile butadiene styrene (ABS) [46,47,48,49,50]. Additives commonly used in the plastic processing includes colorants, bonding agents, UV protective agents or stabilizers, biocides, lubricants, and pigments which help to tailor the end product to a specific area of application [51]. Investigations are ongoing which have focused on facet of compatibility between plastics commonly used to blend wood fiber or flour to elevate performance. Other efforts have been made to lessen the density by foaming the core of the extruded profile or by molding into hollow sections for specific use, refer to authors [52,53,54,55,56]. Moving into investigation of injection molding, an appreciable work has been undertaken to address surface quality and mold filling ability of the WPC melt, as viscosity increases in the presence of the wood flour [57].

Numerous studies focused on the development of WPC production which usually undergoes two different phases, namely: monomers which can be introduced into the wood pores by several techniques depending on the properties of the wood or being treated. This is then followed by their polymerization inside the wood [58]. The end product is similar to natural wood, and its properties are enhanced by the combination of wood and polymer material. Thus, the wood components improve the hardness, abrasion resistance, dimensional stability, compression or bending strength, resistance to biological degradation, etc. [59,60].

### 2.1. Polypropylene (PP) Polymer Composites

Polypropylene (PP) polymer is a thermoplastic polymer that can be made by polymerizing propylene molecules. It is used in composite material manufacturing as a matrix material due to its excellent properties. PP is also very suitable for filling reinforcement and blending which give high rigidity, hardness, low-cost effectiveness with strength and excellent resistance to environmental stress cracking among many others [61,62,63,64,65,66,67]. PP combined with natural fiber polymer is one of the most promising technologies to create natural-synthetic polymer composites. PP composite-based materials are processed by extrusion molding, injection molding and sometimes expansion molding. The end products are used in several applications in the construction or building, automotive, medical, textiles, and packaging industries, etc. In building sector, PP composites is modified to fabricate roofing materials giving an excellent property including thermal stability, UV resistant, acid rain-resistant, light weight, flexible and so on [68].

During the production of PP composites, natural fibers are commonly used as reinforcement to enhance the mechanical and thermal properties of the finished product for excellent performance. There have been several studies on PP-based natural composites, to evaluate their mechanical properties and performance [69,70]. Bledzki and coworkers [71] reported the production of PP blend with wood fiber, wood flakes, wood chips from hard wood, softwood to form composites through injection molding process. The effect of fiber length, structure and coupling agent that was assessed including the mechanical properties of the composite materials. The wood chip was used as reinforcing agent with 5 percent maleic anhydride (MA) to increase the adhesion between the fiber and the matrix. Again, to enhance the dispersion of the particles and to lessen the water sorption properties of the finished composite. Their investigations show that wood chips-PP composites gave a desirable tensile and flexural properties in contrast with other wood fibers blend with PP composites. An increment of strength was realized when MA was added to the wood chips-PP composites showing an increase of 65% to enable the structure to survive impact induced damages.

Another investigation studied by the same authors [72] about the effects of applying chemical foaming agent concentration to attain better mechanical properties, cell morphology and density. Injection mold and extrusion processes were employed to produce wood fiber reinforced polypropylene (PP) composites in this case. The outcome shows that mechanical properties were higher by 80% with the addition of coupling agent to enhance the performance of the composite material. Additionally, the density was reduced when the foaming agent content was added to a maximum of 30% and a drop of about 0.740 g/cm^3^ by injection molding and 0.83 g/cm^3^ decrease when the density was reduced by 20% through extrusion process at wood fiber constituent of 30 wt% mixed with the coupling agent.

Chattopadhyay and others [73] worked on the physical, mechanical, and thermal behavior of bamboo fiber reinforced polypropylene (PP) matrix composites. They employed compression molding process where the fiber volume percentages were measured under 30, 40, 50, and 60 percent. To enhance the fiber-matrix adhesion, maleic anhydride (MA) grafted polypropylene was used as the compatibilizer. Five (5) percent maleic anhydride-grafted polypropylene composites concentration prepared at 50% fiber volume had an impact strength of 37% increment. Flexural strength also increased by 81% while the tensile strength, tensile modulus gained an increment of 105% and 191% respectfully. Additionally, the reinforcement of 50% volume bamboo fiber/polypropylene matrix composite showed an about 23% increment. The mechanical properties and thermal stability were quite high exhibiting an improvement in their reinforced thermoplastic composites.

Kenaf fiber (KF) reinforced polypropylene composites manufacturing were investigated by Asumani et al. [74] by compression molding techniques. The KF was studied in three categories; untreated, treated with NaOH and alkali-silane treatment with varying fiber weight percentage (20%, 25%, 30%, 35%). The effects of these chemicals significantly improved the mechanical properties of the KF polypropylene composites. The tensile and flexural properties were improved, resulting from alkali-silane treatment due to better bonding between the fibers and the matrix showing from scanning electron microscopy.

The same group, Asumani et al. [75], conducted more research on the effects of two fiber treatments, thus; alkaline alone treatment and combination of alkaline-silane treatment on the fatigue and impact strengths of KF reinforced polypropylene composites. The results indicated an improvement on the fatigue and impact strength by fiber treatment, while treatment with alkali-silane treatment was the most effective one. Five to six percent alkali concentration was found to be the optimal concentration for the two treatments (alkali-alone and combine alkali-silane). The results shown by microscopy examination exposed the damaged mechanisms of the composites which were influenced significantly by the fiber treatment. The failure of the alkali-alone treated KF/PP composites was characterized by poor fiber-matrix adhesion, while the failure of alkali-silane treatment of KF/pp composites was characterized by fiber breakage and matrix cracks revealing improved fiber-matrix adhesion due to the additional silane treatment.

Rana et al. [76] fabricated jute fiber (JF) reinforced polypropylene (PP) matrix composites with varying fiber concentration from 20% and 5% intervals to 35% weight percentage using injection molding techniques. However, the fiber loading was from 30 to 60 wt% in 10 weight percent intervals under compatibilizer doses of 0, 1, 2, 3, and 4 weight percent. Jute fiber was oxidized and the fabricated jute fiber/PP matrix composites were post-treated with 15% urea solution. The flexural and tensile strength were improved even with 1% compatibilizer only. The hardness was also increased due to oxidation of jute fiber, post treatment of manufactured composites with urea solution and fiber loading. The increased hardness was due to the improved stiffness of the composites.

Kwaekuk et al. [77] also conducted an experiment to determine the physical properties of sisal fiber reinforced/polypropylene (PP) matrix composites that were formulated at fiber content of varying weight percentages of 10, 20, and 30 wt% using injection molding process. The effects of sisal fiber treatment with alkaline combined with maleic anhydride-grafted propylene was also studied. A better revelation was conceived to ameliorate the mechanical properties of the polypropylene (PP) composites, cellulose decomposition and water resistance of the polypropylene composites. Additionally, the alkali treatment increased the surface area of the PP matrix to improve fiber/matrix adhesion therefore increase the impact strength.

Recently, an investigation was conducted by Das et al. [78] about the peak heat release rate (pHRR) of biochar reinforced polypropylene (PP) matrix bio-composites using injection molding techniques. The sample of the biochar concentration in five varying weight percentages from 0%, 15%, 25%, 30% and 35% weight respectfully. The pHRR of the neat PP in 0% weight recorded 1174 KW/m^2^, nevertheless, when 15% biochar was added, the pHRR decreased to 791 KW/m^2^ which is about 32.6%. Moreover, the pHRR of 25 wt% biochar/PP matrix bio-composites recorded 627.5 KW/m^2^ which decreased by 20.6% compared to 791 KW/m^2^. The decrease in HRR was attributed to the char layer formation induced by the biochar which prevented the heat and mass transfer between the PP and the oxygen.

### 2.2. Polystyrene Composites

Polystyrene (PS) is a thermoplastic polymer made up of many repeating units of styrene monomer usually synthetic aromatic hydrocarbon polymer [79]. It is a thermoplastic polymer which can be softened when heated and may be transformed into a wide range of finished products through manufacturing processes. PP is one of the most widely used plastics in the scale of production due to its excellent surface aesthetics, good stiffness, transparent, hard, and good adhesion to both paint and plating. This makes them an all-important class of polymers for several applications including construction, lab equipment, food packaging etc. [80].

Polystyrene (PS) is commonly available in two forms: crystal polystyrene and high impact polystyrene. PS can be fabricated with ease by involving free radical polymerization of styrene using free radical initiator and the materials may be manufactured by injection molding or extruded. PP is also useful to produce other thermoplastic copolymers including acrylonitrile butadiene styrene (ABS), styrene acrylonitrile (SAN), acrylonitrile styrene acrylate (ASA) and of course to mention foam polystyrene [81,82].

Natural fibers are readily available as a useful reinforcing material for manufacturing polymer-based composites. Composites materials manufactured from natural fiber involving reinforcement with polystyrene matrix exhibit unique functions and improves mechanical properties of the composite materials.

Various studies from [83] were done on wood plastic composites with polystyrene (PS) and pinus radiate as the wood flour. In general, the observation made on the mechanical properties, i.e., the flexural, tensile, and impact properties was improved with wood flour content. Therefore, the tensile modulus obtained a decreasing trend when the wood flour content was higher. However, the Izod impact strength showed the best results with 30 wt% wood flour and 70 wt% recycled polystyrene.

Ratanawilai et al. [84] studied the mechanical properties of rubberwood flour reinforced polystyrene (PS) matrix composites manufactured by using compression molding processes. The fiber samples were weighed under varying weight percentages of 35.8, 45.8, and 55.8%. Additive agents such as maleic anhydride (MA), ultraviolet (UV), and lubricant were used in all the samples. The results show that tensile strength of 35.8 wt% rubberwood/PS matrix composite was 23.12 MPa, while at 45.8 wt%, the tensile strength decreased to 17.8 MPa. The decreased tensile strength with the increased fiber weight may result in high stiffness of the PS matrix and inadequate interfacial adhesion between the particulates of sample matrix. However, the increase in PS matrix improved the esterification between the rubberwood flour/PS matrix composite.

Poletto et al. [85] also studied the mechanical properties of cellulose fiber reinforced polystyrene (PS) matrix composite and the effects of coupling agent under varying fiber weight percent of 10%, 20%, 30 wt%. The composites were produced using extruder and after injection molding. Styrene maleic anhydride of 2 wt% were used as a coupling agent. From the observations, the mechanical properties decreased due to the cellulose fiber loading of more than 20 wt%. Therefore, coupling agent was added to enhance the mechanical properties. Use of the coupling agent greatly improved the storage modulus and decreased the damping peak of the composites because the interfacial adhesion was improved. Moreover, the height of the damping peak was shown to be dependent on the cellulose fiber content and the interfacial adhesion between fiber and the matrix. The addition of 2 wt% styrene maleic anhydride to 10 wt% cellulose fiber/PP matrix composite increased the flexural strength to 66 MPa hence the increment of 63.7%. The adhesion factor proves that involving coupling agent gives better adhesion to enhance the mechanical properties of the composite materials.

Similarly, works by [86] were done to assess the various potentials of expanded polystyrene and wood flour to develop wood plastic composites under varying fiber weight percentages from 0 to 40% using injecting molding techniques. The effects of adding a coupling agent (styrene maleic anhydride) and wood flour loading were also examined to evaluate the mechanical properties of WPC. From the results, it was observed that mechanical properties decrease as the wood flour loading increased without coupling agent. While the flexural modulus of the composite material increased with the addition of the styrene maleic anhydride, hence improved the interfacial bonding between wood flour and the polystyrene matrix.

Furthermore, studies by Singha et al. [87] used compression molding techniques to find the effect of reinforcement on mechanical properties of raw and surface modification of Agave fiber reinforced polystyrene matrix-based composites. The samples were prepared under varying fiber content by weight percentages from 10 to 30 wt%. In their work, graft copolymerization of methyl methacrylate (MMA) and an initiator (ceric ammonium nitrate) was employed for surface modification while the agave fiber was reinforced in the form of particles, short and long fibers with varying dimensions. It was revealed from the studies that 20% fiber content recorded an increment of mechanical properties. It was also found that particle reinforcement portrays optimum mechanical properties than sample which were labeled short and long fiber reinforcement. Those with surface modified fiber reinforced composites, observed to have thermal stability than sample of raw fiber reinforced composites. These sample composites were characterized by FT-IR spectroscopy, SEM, and TGA/DTA experiment.

### 2.3. Polyethylene Composites

Polyethylene (PE) is a thermoplastic polymer of ethylene consisting of long chain monomers that can be attained through polymerization of ethene. PE is one of the most widely used thermoplastics due to its good properties such as high toughness, excellent chemical resistance, low coefficient of friction, electrical insulation properties, ease of processing, etc. [88,89]. The mechanical properties of PE greatly depend on variables including the extent and branching type used, the molecular weight and crystal structure. PE composites have several applications such as packaging, forming sheets, thermal energy storage, biomedical, automotive engineering, etc.

Many studies were done to develop PE composites by employing various additives to improve the interfacial bonding strength. Polyethylene (PE) have varieties which are commonly used classified under different categories based on their densities and branching. The main types of PE are high polyethylene (HDPE), with density 0.95 g/cc, low-density polyethylene (LDPE) also with density approximately between 0.910 g/cc and 0.925 g/cc, linear low-density polyethylene (LLDPE) with density from 0.915 g/cc to 0.930 g/cc [90]. HDPE gives excellent stiffness, rigidity, and actively improves heat resistance, though lightweight yet super strong. Again, HDPE is usually used for making bottles, pipe systems, for packaging, etc., while LLDPE and LDPE are also used for film packaging, electrical insulation, etc.

Investigations made by [91] employed the injection molding process to manufacture 20 wt% and 40 wt% spruce wood flour (WF) reinforced high density polyethylene (HDPE) matrix composite. Again, studied the effect of boron-based flame retardant and wood fiber loading on mechanical, flammability and thermal performance of the matrix composites. The wood flour was treated with boric acid solution, borax decahydrate and their mixture were dried prior to composite production. The results recorded better mechanical properties with boric acid and borax solution indicated 19 percent better progression on tensile modulus for 40 wt% fiber loading contrast to the control samples. Additionally, the decreased impact strength was related to the high brittleness in the HDPE matrix due to the wood flour.

Other studies by [92] fabricated agave, coir and pine fiber reinforced liner medium density polyethylene (LMDPE) with varying fiber weight percentages from 0 wt% to 40 wt% using rotomolding techniques. The process was surface treated with maleate polyethylene to enhance the mechanical properties of the fiber composites and to increase the fiber content in the composites. The tensile strength of untreated 10 wt% agave fiber/LMDPE matrix increased when the pre-treatment was used. This was as a result of better attraction, interfacial adhesion and compatibility between the fiber and the matrix. Agave and coir fiber treatment with maleate polyethylene was successful than pine fiber which was attributed to their various chemical composition. Moreover, surface treatment showed a stronger fiber distribution and steadier composite morphology giving way to employ higher fiber content in rotational molding. At low fiber contents 10 wt% and 20 wt%, the mechanical properties were enhanced with administering treated fiber composites compared to the neat polymer and untreated fiber composites.

Torres et al. [93] manufactured high density polyethylene (HDPE) from natural fiber (sisal, cabuya) composites with varying fiber concentration ranging from 0 wt% to 7.5% by weight using rotomoulding process. They determined the mechanical properties of the natural fiber reinforced and unreinforced samples. Results obtained showed that the mechanical properties of fiber reinforced composites are better compared to unreinforced ones. However, it was also indicated that the best performance was obtained from cabuya fibers at a concentration of 2.5 wt% but sharply dropped at 5 wt% even below that of unreinforced HDPE. Again, sisal fiber lessened the performance of the material at 2.5 wt% but gave an improvement at 5 wt%. It was also discovered that low mechanical properties at high fiber contents are usually due to presence of fiber clumps or bubbles during the rotomoulding process and formation of more voids.

Wang et al. [94] produced linear low-density and high density polyethylene (LLDPE, HDPE) with 10% treated flax fiber to investigate the effects and mechanical performance of the fiber reinforced composites using extrusion techniques. From their observations, it was established that in order to enhance fiber/matrix interfacial properties, fibers must be exposed to additives or chemical treatment such as benzoyl, silane and peroxide treatment. The reason is that lack of better interfacial adhesion and poor resistance to moisture absorption render the use of natural fiber reinforced composites not appealing. Additionally, pre-treatment improved the surface properties of the flax fiber and fiber- matrix adhesion. Therefore, treatment with silane, peroxide and benzoyl improved the mechanical properties compared to the untreated fiber because of the good adhesion between the matrix and the fiber. Lastly, the tensile strength was observed to go higher from 15.2 to 16.1 MPa and the impact strength increasing from 190 to 220 KJ/m^2^.

Jayaraman and co-authors [95] manufactured WPC composites of linear medium density polyethylene (LMDPE) with wood fiber and sisal to determine the tensile and impact properties using rotomoulding processes with varying fiber contents from 5 to 25 wt%. The authors discovered that tensile strength was reduced when the fiber was added from 18 MPa (neat polymer matrix) to 14 and 7.5 MPa in the presence of composites with 20 percent of sisal and wood fiber. The tensile modulus improved from 900 MPa (neat matrix) to an increase of 1050 MPa and 1100 MPa for composites with sisal 20% and wood 15% accordingly. However, the tensile modulus declined to 600 MPa and 800 MPa in the presence of 25% sisal and 20% wood fiber. Again, the impact strength gave a higher value at 25% sisal and 20% wood fiber from 85 (neat matrix) to 112 and 145 J/m. Therefore, the tensile strength observed showed a constant decrease with increase in fiber content reported for both sisal (LMDPE) and wood fiber (LMDPE) composites. Additionally, the tensile moduli increased with increasing fiber content to 15 wt% of sisal and wood fiber LMDPE composites.

Similar studies by the authors Banuelos-Lopez et al. [96] prepared LMDPE composites with agave fiber at 5–15% weight content to find the mechanical properties using compression and extrusion processes. From their investigations, it was reported that the impact strength decreased with fiber content from 7.5 J/m for neat matrix to 2.3 J/m under 15% fiber. However, the tensile modulus increased at 10% fiber content indicated at 435 MPa which was 70% higher than the neat matrix.

Cisneroz-Lopez et al. [97] investigated linear medium density polyethylene (LMPDPE) with agave fiber reinforced employing dry-blending processes to find the young modulus of the fiber. The fiber content was 15 wt% used for the reinforced in the LMDPE matrix while maleic anhydride was used to treat the PE to enhance the fiber/matrix interfacial adhesion. The trend from young modulus indicated that agave fiber/LMDPE composites at 15 wt% increased from 0.192 GPa to 0.217 GPa. This increment may be associated with the reinforcement of the rigid phase of the agave fiber in the LMDPE matrix. This improved young’s modulus led to an enhancement in the stiffness of the composite material.

Zhao et al. [98] studied and reported the effect of manufacturing techniques, fiber content, and interfacial modification on the mechanical properties of sisal fiber reinforced HDPE composites using the blow molding process. The fiber content weight percent ranged from 10 to 30 wt%. The results from their investigations showed that as the fiber loading increases, the tensile strength, tensile modulus, and creep resistance of the composites also rises, hence, the tensile modulus increased by 58%, 150%, and 257%. Mechanical properties of the composites were significantly improved owing to the adaptation of maleic anhydride-grafted HDPE because the interfacial adhesion between the sisal fiber and the PE matrix was enhanced.

Hanana et al. [99] worked on linear low-density polyethylene (LLDPE) composites and maple wood fibers both untreated and treated with coupling agent MA grafted polyethylene (MAPE) using rotomoulding. The process is used to find the influence of employing particle size, fiber content and surface treatment on mechanical properties of the composites. From their work, surface treatment with MAPE notably improved the fiber-matrix interfacial quality. This in tend indicated a better thermal stability, homogeneity and enhanced the mechanical properties of the composites. Additionally, the effect of particle size affected the tensile modulus where it increased by 7, 40 and 73% at 30 wt% of maple fibers while the tensile strength increased by 114% at the same 30 wt% fiber content when the particle size increased. Lastly, the impact strength was improved with LLDPE/MAPE solution modified maple (355–500 mm particle) by 52% higher than particle size of (125–250 mm) at the same fiber loading 30 wt%.

### 2.4. Polyvinyl Chloride (PVC)

PVC is mainly made up of units of vinyl chloride monomers. The monomers are processed by polymerization process to form a polyvinyl chloride where the vinyl is derived from salt and ethylene from natural gas. Polyvinyl chloride (PVC) is one of the abundant and most used thermoplastic polymer consumers used daily due to its versatile applications and characteristics. It has a good corrosive resistance, thermal stability and can also possess inherent flame retardant qualities owing to the presence of chlorine in the polymer matrix.

Before PVC can be manufactured into useful products, it needs to be incorporated with a range of specific additives which have an effect on the product properties. The functional additives employed in PVC products are heat stabilizers, lubricants, and plasticizers. These additives usually influence the mechanical and electrical properties, light, and thermal stability of the PVC products. Some also include impact modifiers, thermal modifiers, UV stabilizers, mineral fillers, pigment, etc. They can be efficiently be fabricated into a wide range of rigid and flexible PVC products depending on the type and amount of plasticizer applied [100,101,102].

PVC is a versatile material which offers diverse applications because of its low cost, and desirable physical and mechanical properties. Such applications are used extensively in areas such as construction, electronics, medical, packaging, automotive, sport, and many others [103,104,105]. The production techniques to manufacture PVC products can be blow molding, compression molding or injection molding. Recently, more attention has been focused on thermoplastic composites such as wood-PVC composites due to its advantages and added properties compared to the individual use of wood [103,106].

Aliyegebenoma at al. [107], reported the mechanical properties of polyvinylchloride-sawdust (PVC-sawdust) composite using injection mold process. It was revealed that the tensile strength, proof strength, flexural modulus of PVC-sawdust composite was maximized with 43.7 MPa, 48.4 MPa, 61.4 MPa and 3.42 GPa at temperature 224.65 °C and 61.5% polymer level.

Similar work by Enayati et al. [108] also concentrated on the prospects of the use of PVC waste and sawdust (cut and edging MDF panels) in wood plastic composite manufacturing. The panel sample used compression mold techniques to determine the physical and mechanical properties of the panel including the measurement of thickness swelling (TS). The test boards were with variable mixing ratio of 45/55%, 50/50% and 40/60% with press time 8, 10 and 12 min. Based on their findings, it was established that mechanical properties of PVC/sawdust were significantly higher when 50/50% ratio was employed. Thickness swelling (TS) in 24 h decreased with the increased PVC/sawdust ratios. Therefore, the physical and mechanical properties of composites was affected by press time. It was concluded that the panel made of 50% PVC mixed with 50% sawdust pressed for 8 min shows effective properties and is suitable for several applications such as panels and pipes.

Qian et al. [109] produced a composite made from 70 wt% moso bamboo particles and polyvinyl chloride (PVC) matrix employing varying hydrothermal temperature from 120–280 °C with 20 °C temperature increments by compression molding process. From their work, it was stated that the tensile strength of 70 wt% moso bamboo/PVC matrix composite treated at 140 °C was found to be 9 MPa. While the treated temperature increased to 180 °C, the tensile strength was improved by 74.4% to 15.7 MPa. However, the treated moso bamboo at 180 °C was observed to show smaller gaps in the interfacial region and fewer hydrophilic groups. The treatment process brought about the removal of hemicellulose, wax, and pectin contributing to the improvement of mechanical interlocking, increased cellulose and lignin redistribution. Hence, causing interfacial adhesion improvement between the moso bamboo/PVC matrix and consequently, enhancement in tensile strength.

Pulngern et al. [110] studied the influence of temperature on mechanical properties and tensile creep responses of wood particles (WP) reinforced polyvinyl chloride (WP/PVC) composite matrix under varying temperature 25, 40, 50, 60, and 70 °C through twin-screw extruder. Wood particle (WP) and PVC compound weight ratio was 1:1, while WP was treated with 1 wt.% silane coupling agent followed by various necessary additives (calcium carbonate, calcium stearate). From their observations, the mechanical strength of WP/PVC composites was noted to decrease at temperatures above 50 °C whereas the modulus of elasticity was significantly affected by the temperature also above 60 °C. The tensile modulus of WP/PVC matrix composite at temperature 40 and 70 °C recorded were 3.4 GPa and 1.2 GPa with a decreased percentage of 64.7. Therefore, it was concluded that temperature directly affects mechanical properties of WP/PVC and they are more susceptible to temperature change compared to flexural modulus at small deformation. Hence, an increase in temperature decreased the tensile modulus and this variation in tensile modulus of the composite material was largely dependent on the temperature.

Abdellah et al. [111] manufactured wood flour (WF) reinforced polyvinyl chloride (PVC) matrix composite by compression molding with precipitated calcium carbonate (PCC) used as fillers. The study was conducted to determine the mechanical properties of WF/PVC/PCC composites. The wood flour weight percentages used were 62%, 52%, 42%, and 31% whereas the PCC weight percentages include 10%, 20%, 31%, and 62%. Tensile modulus of the composites was observed to increased when PCC and WF was added to PVC matrix. This may be associated with restraining effect of filler on polymer molecules. Tensile and flexural strength of WF/PVC composite were enhanced by addition of low content of PCC. This gave a good dispersion of PCC particles in PVC matrix of 10 wt.% than with composite with 62 wt% WF. It was revealed from the test that at higher weight percentages, tensile and flexural strength reduces due to lack of an even distribution of PCC in WF/PVC composite matrix which improves the mechanical properties. Another observation from SEM micrograph revealed that neat PVC compound portrayed a uniform and homogeneous fractured surface while PVC/62 wt% WF indicated several voids representing poor WF/PVC interaction. Load content of 52 wt% and 10 wt% WF/PVC composites showed the best dispersion of PCC in WF/PVC composite matrix. Hence, lower filler content leads to better dispersion of PCC particles inside the matrix.

An analysis was made by Song et al. [112] on the flexural modulus of plywood veneers (PV) reinforced polyvinyl chloride (PVC) film composite matrix with modifier under varying weight percentages of 0%, 1%, 3% and 5% through hot pressing techniques. The plywood veneer surface was treated with a modifier 3-aminopropyl (tri-ethoxy) silane to improve the compatibility with the film. The flexural modulus under 1 wt.% modifier concentration increased to 3 wt% thus showing an increment of 5.2%. Furthermore, the flexural modulus of 5 wt% modifier treated PV/PVC film composite matrix was recorded to be 8.5 GPa with a decreased of 15% compared to 3 wt% modifier treated composite. Treatment of PV with a modifier significantly increased the interfacial adhesion of the composite. Thus, the improved interfacial adhesion enabled an efficient stress transfer which resulted in a higher resistance to flexural fracture and deformations thereby an increased in flexural modulus. However, the decreased flexural modulus was associated with over usage of modifier, weakening the interfacial adhesion of the composite. Overall, the modifier improved the thermal stability of composites.

### 2.5. Polylactic Acid (PLA) Composites

Polylactic acid is a thermoplastic aliphatic polyester made with two monomers or building units, thus lactic acid and lactide. Lactic acid can be manufactured by bacterial fermentation from natural sources such as carbohydrate under controlled temperature. PLA is a biocompatible, renewable and biodegradable polymer obtained mainly from corn starch, sugarcane (sucrose in molasses), cassava roots, potatoes etc. making the process sustainable and economical. The PLA is also a hydrophobic polymer with a slow degradation rate, low thermal stability and poor toughness, although it has good mechanical properties, ease of processing and are readily available. The degradation is frequently due to hydrolysis of ester linkages usually along the backbone of the polymer [113,114].

The production of PLA is possible by extraction of starch, fermentation of sucrose, and breakdown of sucrose followed by two manufacturing techniques (lactide conversion and polymerization). These extraction methods use starch from agricultural products, such as stems, straw, husks, grass, etc. They are used as an alternative carbohydrate source to convert the extracted starch to fermentable sugar (glucose and dextrose through enzymatic hydrolysis). Residues which cannot be easily fermented can be used as a heat source to reduce the use of fossil-derived hydrocarbons.

Production of PLA using lactic acid monomer involves two processes: the conventional process and the ring-opening polymerization process.

Production of PLA by conventional process involves polycondensation/direct condensation of lactic acid. Nevertheless, the outcome of the process results in low molecular weight or less-desirable low-density PLA due to various technicalities of removing water and impurities. Therefore, to form high density PLA, the lactic acid is carried out under high temperature and heated in the presence of an acid catalyst to form cyclic lactide or solvent to separate the water produced during the condensation process. The other process involves ring-opening polymerization of cyclic dimer or in the presence of metal catalysts to form high density PLA or higher molecular weight PLA polymer [115,116].

The production of PLA polymer is quite cheap and has made it possible to use in a wide range of applications, especially in electronics, chemical, biomedical field, manufacturing of fabrics, etc. [115,117,118]. The production techniques for PLA polymer are extrusion, biaxial stretching, spinning, and injection blow molding processes. These techniques are dependent on the type of applications intended to be used for [119,120].

To obtain an excellent mechanical properties of PLA polymer, natural fibers can be used as reinforcement to fabricate composite materials which will improve its performance. Natural fibers such as wood, hemp, leaf fiber, sisal, jute, coir etc., are used as reinforcement materials to PLA polymer due to their availability, ease of production, biodegradability and low cost. Several investigations were made to study the manufacturing and mechanical properties of PLA composites.

Du et al. [121] reported the effect of mechanical and thermal properties of polymer composites that were fabricated from natural fibers (hard wood, soft wood, kraft soft wood, pulp fiber) reinforced polylactic acid (PLA) matrix composites. The natural fibers were characterized with varying fiber weight percentages of 30, 40 and 50 wt% through film-stacking process. The tensile strength of 40 wt% hard wood/PLA matrix composite was observed to be 100 MPa and further decreased to 96 MPa with the reinforced of 50 wt% hard wood showing a decrement of about 4%. Furthermore, the tensile strength decreased with increasing hardwood fiber loading which was as a result of the hardwood with the short fiber length and highest surface area leading to insufficient wetting of hardwood fiber by the PLA polymer. Pulp fiber showed a better modulus and strength on the PLA matrix composite while softwood achieved 50% maximum composite strength. Involvement of pulp fibers appreciably increased the composite storage moduli, elasticity and improved the crystallization of PLA. It was observed that fiber to fiber bond may have a great impact and could contribute to the enhancement in composite strength.

Tan et al. [122] studied the performance of PLA composite using a deep eutectic solvent to treat microscale cellulose (MCC) from pulp fiber and also analyzed the mechanical and thermal properties of the PLA matrix. To obtain a uniform distribution and texture, the cellulose (MCC) was combined with PLA composites by solution mixing, melt-blending, hot pressing and cold pressing methods. With the addition of 0.5–1% cellulose fiber, the tensile strength and tensile modulus of MCC/PLA composite materials increased by 27%. The elongation at break of the neat polylactic acid (PLA) did not show any decrement and was retained. Nonetheless, both thermal stability and total crystallinity of PLA was noted to decrease due to the addition of cellulose. It was therefore concluded that a rough surface and high loading amount of the cellulose (MCC) enhance the tensile properties and storage modulus of the composite.

Bajpai et al. [123] investigated the production of grewia optiva and sisal fiber reinforced polylactic acid (PLA) matrix composite using film-stacking process. A fraction of 20% fiber weight was constantly employed for each composite. The tensile strength of 20 wt% grewia/PLA matrix composite was recorded to be 74 MPa which eventually increased to 81 MPa on the reinforcement of 20 wt% sisal fiber with about 9.5% increment. Comparing the two materials used; the tensile strength of sisal/PLA matrix bio-composite was improved than grewia/PLA matrix bio-composite. Therefore, the interfacial adhesion of sisal/PLA matrix bio-composite was much better owing to the enhanced tensile strength of the bio-composite.

Yu et al. [124] developed short ramie fiber reinforced polylactic acid (PLA) matrix composite with and without maleic anhydride (MA) using hot pressing methods to assess the influence on the mechanical properties of the composite material. The ramie fiber weight percent used as reinforcement was 30 wt%, maleic anhydride (MA) 2.1 wt% was used as a compatibilizer for ramie fiber/PLA composites and in the presence of 0.35 wt% benzoyl peroxide (BP) as the initiator. 30 wt% ramie fiber/PLA matrix composite with MA showed a tensile modulus of 4.4 GPa which subsequently had an increment of 2.33% when compared to 4.3 GPa reinforcement of 30 wt% ramie fiber/PLA matrix composite without MA. Moreover, the addition of maleic anhydride (MA) to the ramie fiber/PLA matrix composite increased the tensile modulus and improved the interfacial adhesion between the fiber and the matrix.

A study was presented by Singh et al. [125] to examine the outcome of Dalbergia sissoo wood waste on mechanical and thermal properties of polylactic acid (PLA)-based composites. The wood waste composites weight percentages were 2.5, 5, 7.5 and 10%, blended with PLA granules using melt compounding. It was reported that an increase of wood waste content caused an increase in modulus, porosity. Involvement of water dropped the tensile strength and impact strength at break. However, the flexural strength recorded had similar results to unfilled PLA composite and continued to be stable regardless of the amount of wood waste applied. Analysis from differential scanning calorimetry (DSC) portrayed an increased glass transition and cold crystallization temperatures of the PLA matrix composite.

Samouh et al. [126] evaluated the influence of sisal fiber content on the mechanical and thermal properties of sisal fiber reinforced polylactic acid (PLA) matrix composite under different fiber weight 5%, 10%, and 15% by extrusion and then injection molding process. The outcome revealed that flexural modulus elevated with an increasing fiber content hence the rate of reinforced sisal fiber content in PLA composites improved the interfacial bonding between the sisal fiber and PLA matrix. It was seen that the flexural modulus of 15 wt% sisal fiber/PLA composite was 5.7 GPa. Eventually it improved by 11.76%. compared to 5.1% GPa on the reinforcement of sisal fiber/PLA under 10 wt% loading. An increase of the sisal fiber content led to an increase in crystallinity of the matrix from 47–61% because sisal fiber behaved as a nucleating agent for the polylactic acid (PLA). The tensile strength, the flexural modulus and impact properties were improved due to the increased sisal fiber content in the PLA.

Among many other investigations carried out by these authors, [127,128,129,130] also employed various techniques to improve the PLA matrix composites materials with natural fiber such as wood fibers, sisal fiber, etc.

## 3. Fire Behavior of Wood-Based Composites

Wood-based composite (WBC) materials are created to provide desirable properties for diversity of applications such as structural or construction field among many others. However, one of the main limitations for the use of WBC materials is the tendency to burn when exposed to fire, their inherent combustibility and the associated risk from fire causes injuries, loss of lives and properties. Although some composites materials such as wood plastic composites (WPC), typically PVC composite materials are self-extinguishing materials; however, due to their toxicity and generation of gases, it creates ecological problems and poses health risks [131]. Other types of plastics generally used in WPC formulation exhibit higher fire hazards than wood alone, as plastic materials have higher chemical heat content and can melt, releasing volatile gases [132,133]. The addition of content of plastic and wood to form composite results in the potential higher fire hazards in wood plastic composites as compared with wood.

In the field of fire safety, wood pyrolysis is of basic importance to consider in structural fire where structural wood plastic composite (WPC) members undergo pyrolysis and subsequent combustion. The overall process is complex and it also depends on the extent of the fire scenario, especially the amount of composites materials used for the products and other fuel source. Many studies of burning behavior of wood have been reported [134,135,136], yet still there are significant gaps remaining in solving the fire behavior or fire hazards of wood plastic composite (WPC). The fire hazard of WPC materials is often defined by their fire reaction which describes their flammability and combustion properties that affect the early stages of fire, often from ignition to flashover.

Generally, most polymer composites materials contain chains of hydrocarbon fuels that drive the growth of a fire. When wood plastic composite element is exposed to fire and is continuously heated at an elevated temperature above 300 to 400 °C, changes begin to take place in its structure, the polymer matrix thermally decomposes to a mixture of volatile gases, tar and carbonaceous char. If it contains organic fibers used as reinforcements for example aramid and polyethylene, they decompose and contribute to the growth of heat, smoke and gases. Composites also soften, creep, and distort when heated to a moderate temperature greater than 100–200 °C and this event results to buckling and failure of load-bearing composite structures [137]. The initial thermal decomposition and the ignition of the composite is driven by the fire. Fire behavior of WPC materials may differ in size, shape and surface profile. Since there is an involvement of wood (wood-based), there may exist some amount of moisture. In the course of combustion, absorbed water evaporates from the wood-based material and influence the burning process. General effect of water on fire behavior is already is investigated by [138,139].

Usually, polymer matrixes and organic fibers decompose over temperature range from 350 to 600 degrees Celsius with generation of flammable gases. Decomposition occurs by the chains into low molecular weight volatiles that diffuse into flame. The volatiles consist of varieties of vapors and gases of both flammables including carbon monoxide. The diffusion generated from decomposing wood-based composites into the flame zone react with oxygen leads to the formation of final combustion products with release of heat [137,140]. The burning decomposition material may give feedback into the main fire rising to an intense heat. Figure 2 shows the reaction mechanism occurring during thermal decomposition of a polymer composites.

The reaction of thermal decomposition of polymer may progress to oxidative processes or by the action of heat. Thick composite materials tend to impede the diffusion of volatiles and slows the reaction of the fire but the surface decomposes and under intense heat decomposes into the inner parts. The decomposition of wood-based composites (wood plastic composites) depends on the wood species and type of plastic burned. For instance, the heating value of polypropylene (PP) is noted to be more than two times that of wood fiber (pine), Δ$h_{c}^{l}$ = 19.2 MJ kg^−1^, polypropylene (PP), Δ$h_{c}^{l}$ = 43.23 MJ kg^−1^. It is worth noting that PP does not form an appreciable residue but decomposes into tiny fragments of polymer chains and indicates an excessive heat of combustion which leads to further growth of the fire by PP products [142,143]. Lastly, the formation of residual char, the quantity of char created by composite material is associated with chemical nature of the polymer matrix and the organic fiber used.

Fire behavior of WBC material has been a major concern over the decade and much effort has been devoted to assessing and investigating the applications of fire-resistant techniques in reducing their fire hazards. Therefore, there is the need to study the fire behavior of WBC using various techniques to measure flammability properties. Flammability studies of WPC gives information on potential safety measures and traditional measures to protect composites from combustion which includes addition of flame retardants to the composite materials to improve their fire performance [144]. Various techniques have been used to measure fire properties and analyze wood plastic composite materials which includes cone calorimetry, Limiting Oxygen Index (LOI), Thermogravimetry Analyzes (TGA), etc.

Fire properties includes heat release rate, heat release capacity, heat of combustion, temperature, time etc. are used in analyzing flammability properties of WPC. Of several many flammability properties, heat release rate is considered to be the most important parameter in quantifying and predicting fire hazards [144,145]. Stark et al. [146] earlier used cone calorimetry test to determine HRR of wood flour blended with polypropylene (PP-HDPE). The trend observed for HRR of the composite polymer revealed that as the content of polymer in the composite increases, average HRR and peak HRR also increases. The highest value obtained for HRR was with composite of 80% polymer. Wood flour blended with 80% HDPE-ME had higher HRR compared to other composite samples used. It was established that composites with more than 60% polymer and specifically 80% polymer recorded significantly higher HRRs. Therefore, composites with poor performance in heat release rate (HRR) or tendency of flame spread should employ an effective flame retardant in the composite material and study their fire performance. Table 2 outlines some polymers and their flammability properties.

## 4. Properties That Make Wood Suitable as a Reinforcement in Composites

Reinforcing materials such as wood (wood fiber/wood flour) provide high level of strength and stiffness to WPC matrix therefore increasing the mechanical properties of the composite. Wood is a natural composite which is porous, fibrous, less expensive, strong and can efficiently perform a function as a reinforcement to the polymer composite. Wood consists of natural polymers such as hemicellulose, cellulose and lignin, although their properties defer from the synthetic polymers of which they are often mixed. Wood is generally composed of hollow, elongated or spindle-shaped sometimes also called fibers/tracheid which are parallelly arranged to each other along the trunk of the tree [149]. The hollow center of the fiber called the Lumen are either filled with complete or partial resins, gums or growth from neighboring cells depending on the wood species. These fibers are connected firmly and cemented together to form the structural element of the wood tissue [150]. The chemical constituent of wood makes it a good candidate as reinforcing agent for polymers.

Separation of fibers from wood is mainly achieved by various mechanical or chemical pulping processes which influence the properties of the fibers. Fibers which produce good reinforced composite are often influenced by the properties of the fiber and amount of ratio used (length-diameter ratio) or the degree of an applied load allowing efficient transmission to the fiber-matrix phase to enhance mechanical characteristics of the WPC. The extent of this wood fiber load transmission is important to achieve the magnitude of the interfacial bond between the wood fiber and the matrix phases [151]. However, wood flour is also used as reinforcement to the polymer composite.

Wood fiber bundles are reduced and milled to fine particles or texture and combined with the polymer composites with some aspect ratio. Even though applying low ratios of the wood flour limit the reinforcing abilities, it does not compromise the mechanical performance [152]. Selection of wood species for wood flour/wood fiber based on their characteristics improves the reinforced composites. Composite and their flammability properties may depend on the matrix polymer, the type of wood fiber and also the interaction between the constituents. Recently, several composites reinforced with natural fibers, glass, graphite fiber or aramid were developed. However, attention was tailored to wood plastic polymers based on their excellent properties such as low water absorption, durable, weather resistant, ease of maintenance, etc.

Flammability studies of natural fibers such as hemp and flax as composites reinforcement was conducted by Kozlowski and colleagues [153]. The results revealed that hemp and flax fiber had lower heat release rate (HRR) than the counterpart sample. Other studies were performed by Helwig et al. [154] on flammability of PP composites with varying flax fiber content of 12.5%, 20%, 30%, and 40%, respectively. An experiment performed with cone calorimetry showed that at 12.5 weight percent of flax fiber content, the pHRR was 35%, a decreased value than that of neat PP. Furthermore, they found out that flax fiber content of 30% and higher resulted in a lower heat release rate (HRR) and mass loss rate (MLR) of the flax fiber/PP composites. Nevertheless, the flax fiber reinforcement lowered the burning time of the composite. When the fiber content increased above 20%, the characteristics of the composites behaved more like a lignocellulosic material. The later investigations of flammability properties of wood fiber/polypropylene (WF/PP) containing 50% pine wood fiber were presented by Borvsiak et al. [155]. They reported that WF/PP composites had a decreased peak heat release rate (pHRR) and mass loss rate (MLR) compared to unfilled PP. However, HRR, MLR, and smoke generation or the toxic gas released were slightly reduced by wood fiber reinforcement in the polypropylene (PP).

## 5. Effects of Fire Retardants on WPC Materials

To assure public safety and satisfy fire safety requirement for WPC materials in construction applications, etc., it is important to investigate issues concerning fire performance. It is worth noting that employing fire retardants (FR) affects the interaction between wood flour (WF) and the polymeric material properties [156]. For the purpose of improving fire performance of wood plastic composites (WPC), several fire retardants (FR) materials or elements are added to the WPC [157]. Compounds that are efficiently used for fire applications may either consist of a single or combined elements such as bromine, phosphorus, and chlorine [156,158].

A combination of FRs is a strategy to further improve the flame retardancy through synergism between different flame retardants considering the quantity to be used so it does not alter the mechanical property of the composite. However, there are several chemicals used as FRs which are either active or additive/non-reactive and also depending on their chemical constituents. FRs can be categorized into inorganic and organic and it further classified under halogenated and halogen-free, although halogenated (FRs) were considered environmentally unfriendly, several solutions were found to engage halogen-free (FR) [158].

It is well recognized that intumescent formulation such as acid source; as catalyst, blowing agents, carbonizing compounds are able to provide fire protection for flammable materials [156,159]. Nevertheless, ammonium polyphosphate (APP), metal hydroxides (MH) are also common flame retardant formulations used to enhance fire performance of WPC materials [160]. Figure 3 gives a general view of the mechanism of an intumescent FR material, in that fire retardant forms a char during combustion and serves as a barrier on the surface, which delays fire growth or fire propagation.

An extensive literature review of theories of flame retardancy for wood was written by Browne [161]. The theories for the flame retardant of wood are as follows (i) to decrease the flow of heat to restrict the development of combustion, (ii) extinguish the flame or (iii) change the thermal degradation process. Although there are some variations in flame retardant mechanism of wood, in that some may work by creating an intumescent coating or foam that serves as an insulation to the wood surface.

Other flame retardants may in effect elevate the thermal conductivity of the wood in such a way that heat is swiftly dissipated, hindering ignition. Some other theories state that the flame retardant may experience a highly endothermic reaction that draw up enough heat to prevent the surface of the material from burning. Additionally, the theories proposed for quenching flame indicates that flame retardants discharges radicals at pyrolytic temperatures that scavenge hydrogen and hydroxyl radicals from the combustion gas mixture, impeding the spread of the flaming combustion.

In the course of combustion, formation of char by flame retardant in the presence of fuel with non-pyrolytic char tend to prevent fuel release and provide thermal insulation to the base of the wood composite by creating a protective char layer [161]. The FR causes the charring on the wood surface through dehydration of the FR to form double bond in the wood composite. The process of forming charring or carbon layer contributes to the flame retardant (FR) effect. The formation of the char layer acts as a protective boundary, making the char enhance it stability by the decomposing the wood-based composite during the combustion [162]. The char layer also delays the rate of heat transfer from the wood flame and hindering the fire cycle and therefore decreasing the heat release rate and total heat release.

An example is the formulation of intumescent, APP or MH acts as a protective barrier; providing protection against heat transfer to the WPC material which tend to prevent the release of volatile gases [163,164]. Moreover, some materials can form self-carbonaceous residue and also reduce fire load by storing combustible materials. The efficiency of flame retardant (FR) also depends on the constituent of the char and the decomposition products during combustion process [165].

Hence, FR formulation with WPC materials improves or enhances their flammability properties. Table 3 shows some WPC with different FRs, their manufacturing processes and their effect on the WPC.

### 5.1. Halogenated Flame Retardant (HFR)

Chemical element of halogenated FRs is grouped into bromine, chlorine, fluorine and iodine but bromine and chlorine-based halogenated FRs are commonly used due to their effectiveness in flame retardancy. Although there are major environmental concerns brominated-based FRs being more effective are still widely used especially in polyolefins. HFRs directly acts on the core of the fire, i.e., act in the condensed phase, hindering or interfering the chemical reaction involved during combustion [171].

### 5.2. Bromine-Based

Bromine-based flame retardants directly act on fire cycle in the gas phase to break the chemical reaction chain either to prevent the fire propagation or slow it down. Brominated FRs are further classified into three subgroups based on how these compounds are integrated into polymers that is brominated monomers, additive and reactive. Brominated monomer which includes brominated styrene (BS) or brominated butadiene (BB) is usually used to manufacture brominated polymers which are combined with nonhalogenated polymers are included in the pre-polymerization feeding mixture. This process results in a polymer comprising of both brominated and nonbrominated monomers. Additive flame retardant (FRs) such as polybrominated diphenyl ethers (PBDE) and hexabromocylododecane (HBCDD) mixed with polymers during manufacturing. As an additive, it is more likely to leach out from its incorporated polymer matrix into the surrounding environment [172,173]. While reactive FRs such as tetra-bromobisphenol A (TBBPA) are used to chemically bond into the polymer. In other words, bromine base FRs can practically be used with an antimony synergist usually antimony trioxide. Using antimony compound alone is less effect but combining with other halogens form antimony trihalide [174].

### 5.3. Halogen-Free Flame Retardant (HFFR)

The concern for searching for an alternative flame retardant which will perform well as the other FRs but with less cost effects have been a recent focus in research. The focus is to develop and promote a safer environment while using FR products effectively. A wide variety of HFFRs which includes phosphorus, nitrogen, boron and metal-based are commonly used. Some of them have been combined with other FRs to improved flame retardancy. HFFRs require an application with high loading amounts which could have an effect on the mechanical properties of the WPC products [174].

### 5.4. Nitrogen-Based

Nitrogen compounds containing FRs are mainly based on application of pure melamine, melamine homologs or melamine derivatives. Examples are melamine for polyurethane flexible foams, melamine phosphates in polyolefins, melamine phosphate or dicyandiamide and so on. It is gaining rapid growth in FR industries due to it low toxicity and alos a known environmentally friendly FR. Flame retardants based on nitrogen compounds are highly recommended for recycling of many plastics because the nitrogen FR possess high decomposition temperatures. The mechanism at elevated temperature enables the formation of cross-linked structures which aid char formation [175].

Inert nitrogen gases released displays chain reactions by diluting combustion gases. Furthermore, synergistic effects can be attained by integrating phosphorous and nitrogen, where nitrogen improves the coupling of phosphorus to the polymer material. Hence, FRs of nitrogen-based are effective to use as insulators, foams, and other electrical appliances.

### 5.5. Phosphorus-Based

Phosphorous compounds containing FRs extends widely in a range of organic and inorganic compounds including phosphate esters, phosphonates, red phosphorus, phosphinates, which possess good fire safety performance [176]. In fire scenario, flame retardant containing phosphorus releases phosphoric acid which tend to break the combustion process by aiding the char formation owing to incomplete combustion. It has been reported that phosphorus alone does not increase char especially in polyolefins unless there is an introduction of another char forming additives, typically nitrogen containing compounds [177]. Phosphorus-based FRs are widely used in engineering applications such as building and automotive industries. To achieve a phosphorous-nitrogen synergism, melamine is usually coupled with phosphates to promote char formation while decomposing to generate nitrogen and ammonia. These classes of phosphorus-based FR materials are ammonia phosphate (APP), diammonium phosphate (DAPP), and melamine phosphate (MP). They act as intumescent FR which reacts in condensed phase, interfering and restricting combustion of the polymer at its initial stage.

It is worthy to note that, the amount of phosphorus plays a remarkable role in the fire resistance of the phosphorus flame retardant. High content of phosphorus give rise in low flame retardant loading [178]. Moreover, APP is found to be more suitable and effective FR in wood plastic composites. In the work of Arao et al. [179], evaluation of MP and APP was investigated on WPC. They found that blending APP and wood flour (WF) displayed a better result for FR than the FR counterpart MP. An amount of 10 wt.% APP combined with 50 wt.% of WF into polypropylene (PP) is able to produce self-extinguishing properties, achieving UL94 V-0 class. This was attainable due to the fact that APP interact with the carbonaceous structure of wood during combustion which facilitated the generation of char residue. APP aid in synergetic effect of formation of char layer of WPC and non-flammable gas, therefore improving the fire performance of the composite.

Li et al. [180] introduced melamine polyphosphate (MPP) and aluminum hypophosphite (AHP) to evaluate their FR and thermal degradation of WF/HDPE using PE grafted with maleic anhydride. They reported that WF/HDPE composite of 35 wt.% MPP/AHP with 3:2 ratio could attain LOI value of 29.6% and V-0 class of UL-94. Cone results obtained a lower HRR and smoke production while TGA results shows that engaging MPP and AHP enhanced the thermal stability of the WF/HDPE.

However, it was reported by Arne and co-workers [181], who tested the effectiveness of various FRs, APP and other formulations based on phosphorus-nitrogen compounds for the polymeric matrix. Flame retardant compounds were compounded with wood flour, HDPE, and PP to form WPC using injection molding and extrusion processes. Results for LOI shows that; APP obtained 66–67%, phosphorus-nitrogen 65–68% while untreated wood flour (pine) displayed 21–22%. It was also reported that zinc borate displayed the lowest LOI value of 27–28% when 20% FR was applied to all the FR compounds.

### 5.6. Boron-Based

Flame retardancy of boron-based compounds including zinc borates, ammonium pentaborate, melamine borate, metal borophosphate, borosiloxane, and boron phosphate are well known FRs in plastic and rubber industries in recent times. Boron-based FRs are often used owing to their unique properties which function efficiently in smoke suppression, afterglow suppression. Also an agent for antitracking for halogen and halogen—free polymers [182]. Boron-based FRs reacts with hydroxyl containing polymers from cellulose and hemicellulose in order to promote char formation in WPC materials. Of all the FR compounds containing boron, zinc borate (ZB) is most commonly used due to its high thermal stability. It also facilitates the formation of glass esters that forms a protective layer and reduces polymer degradation. Hence, they promote cross-linking between the polymer chains decreasing decomposition of the polymer into volatile flammable gases in (both smoke production and flaming combustion). ZB promotes the reaction with hydrogen halide to create zinc chloride, zinc hydroxychlroride, boric acid and water [183].

Altuntas et al. [184] investigated synergic effects of different flame retardant compounds and used zinc borate (ZB) with polypropylene (PP) and medium density fiberboard (MDF) to produce wood plastic composites (WPC) using the twin screw extrusion process. To further determine the synergic effect of the composite, antimony trioxide (AT), magnesium hydroxide (MH), ammonium phosphate was used. Again, a coupling agent such as maleic anhydride-grafted polypropylene (MAPP) was incorporated during the preparation and mixing of the synergic compound. The reason for adding MAPP was to bring about compatibility in the composite (wood and the plastic). From their observations, the fire resistance of the wood plastic composite had a synergic effect on the burning rate from 16 to 32 mm/min. The composite PP-ZB-AP-AT had the least burning rate of 15.5%, PP-ZB-MH recording a decreased burning rate of 35%, PP-ZB-AT recording 44% while PP-ZB-AP decreased to 55%. LOI results of the WPC showed that PP-ZB-AP-AT, PP-ZB-AP and PP-ZB-AT obtained the highest values of 22% this is because, the synergic compounds used in the studies had greater effect on the LOI results.

The authors [185] produced the composites by extrusion and studied the synergic effects of incorporating ZB and wood flour (poplar) with polyvinyl chloride (PVC) on flame retardancy, thermal decomposition, and other properties. The particle size of the wood flour was from 50–80 mesh size and ZB amount in the WF/PVC/ZB matrix was 6 wt.% according to the total weight of the materials. They stated that ZB had no effect on the FR of the composites but rather reduced the smoke extinction area. The total smoke production was about 43.36% and 50.77% showing ZB as an excellent smoke suppressant of the composite material and also decreasing the production of carbon monoxide by 20%.

### 5.7. Carbon-Based

Carbon-based flame retardant materials are considered to be one of the most remarkable and promising multifunctional flame retardants due to their thermal conductivity, electrical, chemical, and mechanical properties [186]. Flame retardant containing carbon-based nano fillers/nanomaterials that exhibited high efficiency of flame retardancy includes fullerene, graphene, graphene nanosheets, carbon nanotube (CNT), graphene quantum dots, expandable graphite (EG) or carbon black, extensively used in polymer composites recently. However, to achieve an optimum effect of flame retardancy in WPC, the use of synergists such as ammonium phosphate or zinc borate is often dominant. Expandable graphite has been found to be most promising FR with WPC. Nonetheless, it has a limitation of expansion which results in char cracking during burning, poor interfacial adhesion and dispersion. This makes the interior material to be exposed or split therefore hindering the overall effectiveness of the FR properties on the WPC materials [187].

To overcome these situations, expandable graphite (EG) with combination of other FR compounds will provide reinforced char and surface modification to produce a synergetic system resulting in an effective flame retarding material. A combination of EG and APP in WPC modified with coupling agent 3-(methylacryloxyl) propyltrimethoxy silane as FR in wood flour (filler) and polypropylene matrix composites (WF/PP) was studied by Gou et al. [188]. A mixture of 15 wt.% and 25 wt% EG/APP incorporated in the WPC with different ratios were used. When FR content of 25 wt.% was used, LOI values of WPC/APP and WPC/EG was 30.7% and 37.9%. A ratio of 1:1 EG/APP in WPC was observed to be most appropriate content to achieve the best synergistic effect with high LOI value of 39.3%.

Furthermore, adding EG/APP to WPC resulted in an effective flame retardancy, therefore the composite showed a low HRR, THR, and increased char residue. This result was associated with the formation of carbonaceous layer that acted as an insulator and extinguished the fire. Report from Zhao et al. [189], revealed that the combination of aluminum hypophosphite (AP) and melamine cyanurate (MCA) as FR in PP/WF composite resulted in both gas phase and condensed phase flame retardancy during combustion. When polypropylene and wood flour (PP/WF) composite was loaded with 20% AP/MCA under a ratio of 5:1, LOI was 29.5% and the flammability rating achieved UL-94 V-0. The pHRR of WPC with AP/MCA were notably reduced to 304.1 kW m^−2^ which showed the synergist effect of AP/MCA.

### 5.8. Silicon-Based

Silicon-based flame retardants are potential FRs that exhibit greater thermal stability and good resistance with negligible amount of emission of toxic gases during decomposition of polymers. They produce a protective surface coating during fire causing a low rate of heat release and low release of external heat flux in the gas phase. Silicon also shows a lower burning rate which is due to the buildup of silicon ash layer on the silicon fuel surface restricting the diffusion of fuels into the combustion zone and restricting oxygen into the unburned fuel [190,191]. Silicon-based FRs offer significant advantages for FR applications owing to these fire properties including insulation.

The FR mechanism of silicon during combustion forms an inorganic blockade on the surface that protect the underlying polymer from oxygen and prevent the movement or reduce the heat transfer. Under high temperatures, unlike organic polymers, silicones leave behind an inorganic silica residue. The silica residue acts as a shielding effect or an insulation blanket which provide a barrier slowing down the volatilization of decomposition product [191]. Hence, it reduces the rate of volatiles produced for burning in the gas phase and the amount of heat that gives feedback to the polymer surface. Therefore, avoiding the ring formation and increase the formation of silica as the pyrolysis products. Several studies reported that silicone FRs can be used in combination with others as synergists to modify the FR for excellent improvements in flame retardancy in the polymer matrix. However, silicone-based FRs can be enhanced by addition of fillers or additives to improve the flame retardancy which includes; silica, calcium carbonate, wollastonite, mica, talc, carbon black, or some compounded silicone such as polydimethylsiloxane-type which contains dry powders with versatile organic plastics in PS etc. Additionally, silicon derivatives, polycarbonate (PC) offers an excellent mechanical properties and effective flame retardancy performance [192].

Recently, introduction of nano silicon dioxide (nano-SiO_2_) has been combined with other FRs in WPC to generate a synergistic system for improved flame retardancy. The effect of nano-SiO_2_ on WPC (wood fiber/high density polyethylene- WF/HDPE) composite combined with APP were investigated by Pan and co-workers [193] to enhance flame retardancy by synergistic system. The composite was prepared with 70 wt.% HDPE, 30 wt.% WF, nano-SiO_2_ content of 2, 4, 8 and 10 wt.% respectively using a twin-screw extruder. From the cone analysis, addition of nano-SiO_2_ greatly affected HRR, THR and TSR of the WPC composite. An increased nano-SiO_2_ content from 4 wt.% to 8 wt.% resulted in a decreased HRR of WF/HDPE composites from 175.3 kW/m^2^ to 157.0 kW/m^2^ representing about 28% reduction of composites material with no nano-SiO_2_ content. Additionally, a combination of 8% APP, 6% nano-SiO_2_ resulted in a decreased pHRR of 42 and 44% while TTI had an increment of 78% which was attributed to an excellent diffusion of nano-SiO_2_ and better interfacial bonding between the WPC composites. The application of APP and nano-SiO_2_ influenced the early thermal degradation of WF/HDPE and stabilized the residual char formation, improving its physical characteristics and hence giving an effective synergistic effect on the flame retardancy of the WPC.

### 5.9. Other Flame Retardant

Among other FRs are metallic hydroxides typically aluminum-based and magnesium-based, most commonly used as an effective FR in WPC due to their flame retardancy properties. They are low cost, easily produced with other polymers, safe to use due to their nontoxic smoke production and very effective smoke suppressants. This makes them a good candidate to be used for several considerable applications in FR for different polymer matrixes. The metallic FR reaction mechanism is based on hydrated minerals which releases water molecules as they decompose and tend to dilute the combustibles gases by cooling and providing an endothermic reaction [194]. Using this process a cooling effect will give rise to a self-extinguishing ability in the composite material.

The most common halogen free FRs that goes through endothermic reactions and hinder with the combustion process when exposed to flame zone are aluminum tri-hydroxide (ATH) and magnesium hydroxide (MH), or magnesium dihydroxides (MDH). The derivatives of metal oxides as a decomposition product create a nonflammable protective layer on the surface of the composite material. However, employment of ATH and MH have a setback, they have low thermal stability as a reaction can take place at the temperatures of 200 °C (ATH) and 300 °C (MDH). It was also found that they require high loading content usually 50 to 65 wt.% of the filler incorporated in the polymer matrix to achieve a better flame retardant property [195]. This application is subject to the mass of the composite material. However, the loading may differ from ATH and MH application into the polymer matrix.

Katančić and coworkers [196], studied the manufacturing process and the effect of employing aluminum hydroxide (ATH) with ammonium polyphosphate (APP) as fire retardants and other nanofillers to HDPE/WF composite by extrusion process. The WPC ratio of 70:30 and 20% mass concentration were used for ATH and APP. The nanofillers were incorporated to modify the surface of the polymer to enhance the mechanical and fire performance. From their observations, ATH FRs showed a higher thermal stability for the wood composite compared to APP FR. This was due to the fact that ATH acts in endothermic decomposition and releases water vapor while APP functioned as a scavenger of radicals by reducing their concentration. From the results, LOI for ATH is 22%; however, ATH FRs with silica nanofiller showed a higher resistant to fire due to a crystalline structure of the polymer matrix that extended the barrier effect. From the results, some synergistic effects were achieved due to the formation of silica barrier acting as a protective layer during combustion on the WPC surface portraying a thermal insulation layer.

Organoclays are also regarded as worthy FR materials and reinforcement for WPC due to its exceptional properties such as high surface area, high aspect ratio which are very effective for promoting better flame retardancy [197]. Clay reinforcement of the polymeric material is found to give a significant decrease of flammability properties of WPC materials. Hence the presence of clay decreases the burning rate of the WPC products as a result of the formation of carbonaceous layer that is found to protect the materials during combustion [198]. The flame retarding mechanism involves a high-performance of carbonaceous silicate char which reinforced the surface during combustion and acts as an insulator underneath the material to delay the MLR of the decomposition materials [199,200].

## 6. Conclusions and Future Prospects

The capability of wood-based composites (WBC) to be converted to a wide variety of applications in terms of their physical and mechanical strength, affordability, and sustainability, makes them a viable candidate as a solution in lessening the need for solid wood and other traditional materials. Indeed, underused wood wastes and plastics have now been increased in various areas of applications due to their outstanding properties allowing them to successfully replace the demand for solid wood. The global market for WPC is growing fast owing to its potential advantages of economic and material development. Also, composite industries can adapt innovation processing techniques providing manufacturers a versatile adaptability to accomplish broader use of WBC.

WPCs are the future trend and one of the most important future technologies in wood industries. It has a high thriving potential for technical substitutions for construction and building sector expanding into existing and new applications, thus the future looks very successful.

The present review provides a comprehensive guide and opportunities for advanced research on flammability properties and WPCs containing flame retardants. Numerous studies carried out revealed further new challenges, approaches and ideas emerging allowing for captivating and thriving research in the future. It has become ostensible that the manufacturing process, type of flame retardants and polymer matrixes plays an important role in the overall efficiency of flame retardancy, mechanical properties and flammability of WPCs.

The preparation methods and wood fiber/flour content are selected in such a manner that FRs WPC would not compromise with the mechanical property. An appropriate content ratio selection was observed to be the best option to achieve good synergistic effect with high LOI values and low heat release rate (HRR). For instance, 1:1 ratio of graphite with ammonium phosphate showed the most appropriate content to gain an increased LOI value of 39.3%.

It was noted that the introduction of increased additives to FRs in WPCs to generate a synergistic system greatly improves flame retardancy. It became obvious that ammonium phosphate (APP), graphite, and metal hydroxide derivatives were commonly used FRs due to their outstanding flame retardant efficiency.

## Figures and Tables

**Figure 1 polymers-13-04352-f001:**
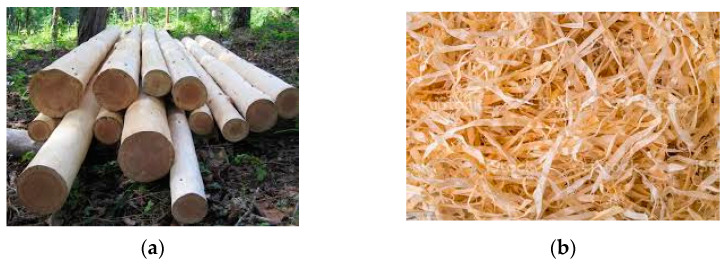
Basic wood elements for wood composites [18], (**a**) Logs, (**b**) Wood fiber, (**c**) Wood Flakes, (**d**) Wood chips, (**e**) Wood powder, (**f**) Wood swiths.

**Figure 2 polymers-13-04352-f002:**
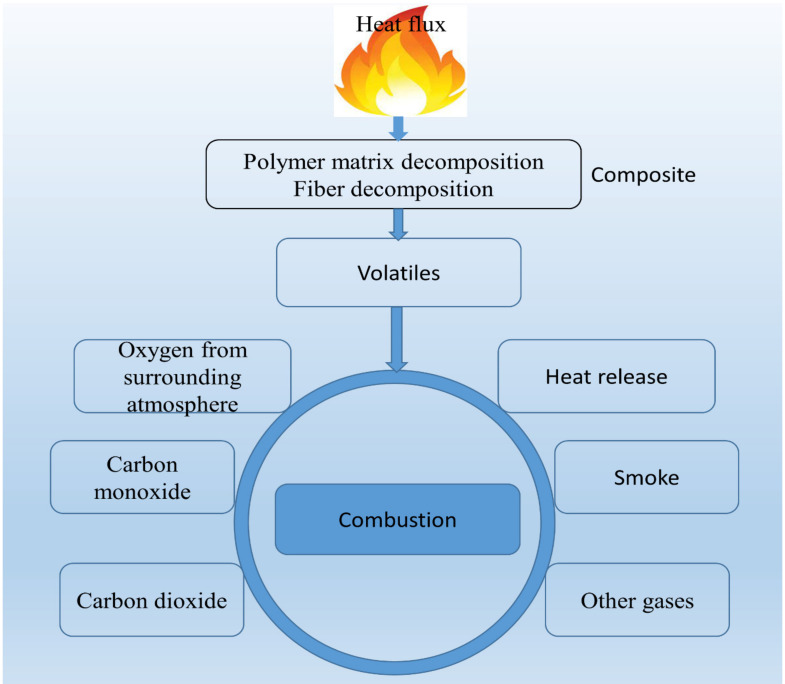
Reaction mechanism process in thermal decomposition of a polymer composites [141].

**Figure 3 polymers-13-04352-f003:**
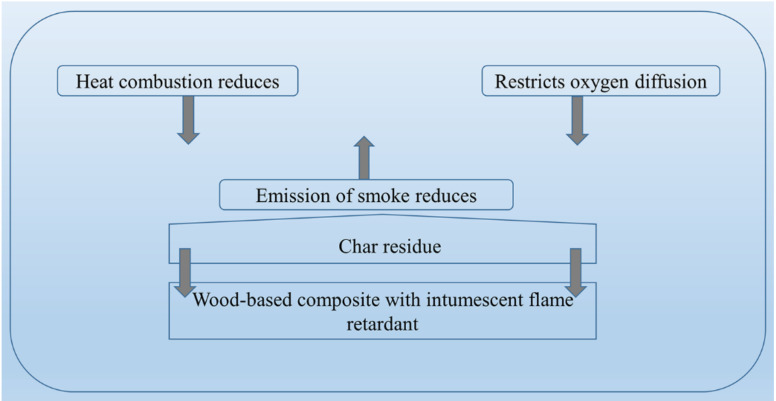
Intumescent flame retardant mechanism (reprint from [162]).

**Table 1 polymers-13-04352-t001:** Types of composites.

Polymer Matrix Composites	Metal Matrix Composites	Ceramic Matrix Composites
Thermosets and Thermoplastics Organic fiber reinforced Glass fiber reinforced Carbon fiber reinforced	Particles reinforced Whiskers reinforced Sheet reinforced	Short fiber composites Long fiber composites

**Table 2 polymers-13-04352-t002:** List of polymers and their flammability properties [147,148].

Polymer (Plastics)	Heat Release Rate (HRR) W/cm^2^	Limiting Oxygen Index (LOI) vol.%
Polypropylene (PP)	150.9	17–18
Polystyrene (PS)	110.1	18
Polyethylene (PE)	140.8	17–18
Polyvinyl chloride (PVC)	17.5	23–45
Polylactic Acid (PLA)	27.2	21

**Table 3 polymers-13-04352-t003:** WPC with various manufacturing processes and their effective flame retardants [145,166,167,168,169,170].

Formulation	Method of Manufacturing Process	Outcome of the Effect of Fire Retardant on WPC Materials	Ref.
WF/PP(MAPP)/ATH/Zinc borate/graphite/T_i_O_2_	Twin-screw extruder	The effects of FRs on fire scenario of WPC shows that the FRS ATH, ZB and melamine had a great impact on the pHRR which significantly decreased by 8 to 22%.	[145]
WF/PE/MH WF/PE/ZB WF/PE/MP	Twin-Screw Extruder	All fire retardant presented a significant improvement of pHRR and average HRR of WPC. WPC/MH had an excellent performance than the other FRs. However, LOI had n increment of 29%, although WPC/ZB also performed well but the result was not different from WPC/MP.	[166]
WF/HDPE/PEC 15% WF/HDPE/APP 15%	Compression Mold	Flame retardant of WPC/APP increased the LOI value by 23.9%, while WPC/PEC was 24.4% which indicate that PEC can enhance the LOI of WPC better than APP. However, a 25% addition of PEC to WPC obtained a higher LOI of 28.7% which represent a UL-94 V-0 rating.	[167]
PP/WF (50/50) PP/WF/AHP (35/35/30) PP/WF/TPP (35/35/30)	Twin screw extruder	PP/WF composite without fire retardant recorded 28.7% mm/min UL-94 completely burnt the sample with low LOI value of 18%. Employing an addition of AHP and TPP fulfilled the UL-94 HB burning rate of 20.8 mm/min and 11.5 mm/min with LOI values of 19% and 21% under 30 wt.% for both AHP and TPP loading. Incorporation of AHP and TPP improved the fire retardant of PP/WF composite.	[168]
PLA/WF/PEG (80/10/10) PLA/OWF/PEG (80/10/10) PLA (100)/APP PLA/OWF/PEG (80/10/10)/APP	Melt-compounding and hot-compression	Blending oxidized wood flour (OWF) and ammonium polyphosphate (APP) to the biocomposite resulted an excellent fire-retardant performance. Peak heat release rate (pHRR) showed a significant reduction, improving LOI by 30.6% and achieving UL-94 V-0 rating standard also.	[169]
PP/WF (60/40) PP/WF/APP (42/28/30) PP/WF/APP/MAPP-5% (37/28/30/5) PP/WF/APP/MAPP-10% (32/28/30/10) PP/WF/APP/MAPP-15% (27/28/30/15)	CO- Rotating Twin Screw Extruder	LOI value of (PP/WF/APP) increased by 17.7% due to the addition of APP from 20.9 (PP/WF) to 24.6 (PP/WF/APP), revealing that APP had a great influenced on the composite with good fire-retardant performance. There was a further increment of LOI value with the incorporation of MAPP. However, PP/WF/APP/MAPP-10% had a higher LOI value of 25.1 indicating 2% higher compared to (PP/WF/APP). Furthermore, PP/WF/APP/MAPP -10% also showed the lowest pHRR of 546.5 kW/m^2^, longest duration of ignition time (IT) and the highest residual mass of 24.2 wt% among all the wood plastic composites. Therefore, the addition of MAPP could maximize the fire-retardant effect in the wood plastic composites.	[170]

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
