# Peer review of "Fire Behavior of Wood-Based Composite Materials"

_polymers, 2021, doi:10.3390/polym13244352_

Round 1
Reviewer 1 Report
Тhe topic of the manuscript is interesting, and the data are well presented. In my opinion, the analysis and assessment given by the authors should be deepened.
In my opinion, in the abstract, the authors should emphasize the main findings of the manuscript.
Please consider modifying the title “3. Challenges: Fire behaviour of wood-based composites” (line 699) to “Fire behaviour of wood-based composites” or something similar.
Please consider combining title 6 “Summary - future prospect and recommendation” and title 7 “Conclusions”. In my opinion, the conclusions should be more concrete and provide the essence of conducted analyses.
The whole manuscript seems a little bit too long, so please consider removing some well-known facts.
The references cited are appropriate.
Author Response
Kindly see the attached response.

Reviewer 2 Report
The paper is well written and structured. Contains comprehensive information. The minor comment to enhance the quality of the present review paper:
- More up to date literature sources of 2021 need to be provided. The majority references are from the previos years.
Author Response
Kindly see the attached response.
